# The Effects of Internal Erosion on the Physical and Mechanical Properties of Tailings under Heavy Rainfall Infiltration

**Rong Gui** [1,2] **and Guicheng He** [1,*]

1   School of Resource & Environment and Safety Engineering, University of South China,
    Hengyang 421001, China; guirong606@163.com
2   Geotechnical Institute, TU Bergakademie Freiberg, 09599 Freiberg, Germany
*   Correspondence: hegc9210@163.com

**Abstract:** The stability of tailings dam will be affected by the internal erosion under unsteady seepage caused by heavy rainfall infiltration which changes the physical and mechanical properties of tailings. In this paper, the hydraulic sedimentary model was established to investigate the effects of dry beach slope on the sedimentary characteristics of tailings in upstream tailings dam, and the results indicated that the dry beach with a larger slope has a more obvious stratification of tailings. Additionally, the sand column model was built to investigate the effects of internal erosion on the physical and mechanical properties of sedimentary tailings under unsteady seepage, and the results indicated that the migration of fine-grained tailings was caused by internal erosion increases the permeability and reduces the shear strength of the tailings. After internal erosion of tailings under heavy rainfall in 50 years return period for 24 h, the average particle size of downstream tailings (sample DT), midstream tailings (sample MT), and upstream tailings (sample UT) increased by 6.4%, 12.0%, and 2.4%, respectively, the hydraulic conductivity of the samples DT, MT, and UT increased by 27.2%, 17.9%, and 15.3%, respectively, and the shear strength of each samples decreased by 20.9%, 15.1%, and 12.4%, respectively.

**Keywords:** upstream tailings dam; sedimentary characteristics; unsteady seepage; internal erosion; particle size distribution; hydraulic conductivity





## 1. Introduction

As the third-largest mining country, China produces about 300 million tons of tailings every year. Additionally, most of the tailings were stored in more than 2000 tailings reservoirs. However, more than 20% of these hydraulic earth structures are at risk of dam failure, which has become one of the most dangerous sources for mining enterprises [1,2]. Serious tailings dam failure accidents have happened all over the world. For example, tailings dam-failure accidents of Xinta Mining Company (Shanxi Province, China) in September 2008 claimed the deaths of 270 people and the direct economic loss of 96.19 million RMB Yuan [3]. On 25 January 2019, the collapse of Córrego do Feijão tailing dam at Brumadinho city (Minas Gerais, Brazil) caused at least 12 million cubic meters of tailing to spread into Paraopeba River and the surrounding area, leaving over 250 people dead [4].

The United States Committee on Large Dams (USCOLD) evaluated the causes of dam-failure accidents, indicating that heavy rainfall was the critical factor that contributes to 30% of the dam-failure accidents [5–7]. When heavy rainfall infiltrates into the tailings, the saturation line of the tailings dam will rise rapidly, as shown in Figure 1. The rise of saturation line increases the hydrostatic pressure and the hydrodynamic pressure of tailings dam, which promotes seepage erosion and affects tailings' physical and mechanical properties. As a widely used dam construction method in China, the upstream tailings dam is more susceptible to internal erosion under heavy rainfall infiltration due to the fine-grained tailings located in the upstream of the seepage direction. The fine-grained tailings are easily carried away from the pore-channel of the coarse-grained tailings by

seepage forces. Therefore, it is necessary to investigate the erosion mechanism and the effects of internal erosion on tailings' physical and mechanical properties under heavy rainfall infiltration.

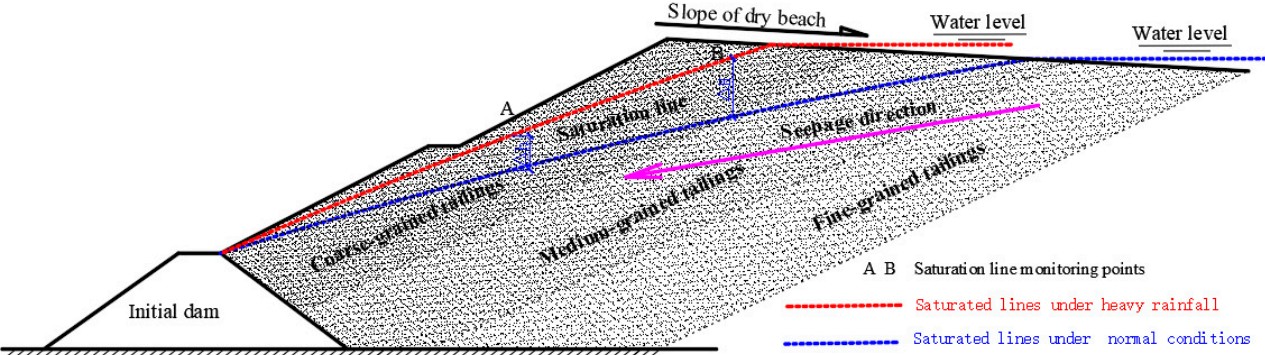

**Figure 1.** Sedimentary and seepage characteristics of upstream tailings dam.

For the past few years, numerous investigations have been conducted concerned with the effect of internal erosion on the soils. Chen et al. [8] investigated the migration trend of the fine particles in cohesionless soils and the change law of content of the remaining fine particles during the process of internal erosion. Zhang [9] established the numerical slope model to analyze the effects of internal erosion on the stability of the slope. Yao et al. [10] investigated the internal erosion mechanism of unstable embankment and analyzed the effect of the increasing water level and hydraulic conductivity on the stability of the embankment. Zhang et al. [11] proposed the prediction formula of the gravel soil's critical vertical upward hydraulic gradient based on the force balance theory of a particle group under internal erosion. Wilson et al. [12] investigated the impact of soil properties on internal erosion and built the streambank stability model to demonstrate the increase in a bank's instability due to undercutting by internal erosion. Chu-Agor et al. [13] developed an empirical sediment transport function to predict internal erosion and undercutting of cohesive bank with time-based on three-dimensional soil block experiments. Ke et al. [14] performed a series of one-dimensional upward seepage tests at a constant water head to cause internal erosion in a soil sample and pointed out that internal erosion causes a significant increase in void ratio and hydraulic conductivity, resulting in a reduction in soil strength. Midgley et al. [15] performed seepage research on a streambank of Dry Creek, and identified that internal erosion plays an important role in the streambank failure, especially when acting in concert with fluvial erosion processes. Jiang et al. [16] used the MICP technology to control the internal erosion of sand-clay mixtures by test apparatus in the laboratory and pointed out that MICP treatment facilitates the reduction in erosion and volumetric contraction of sand–clay mixtures. Liu et al. [17] designed a stress-controlled seepage test apparatus to study the failure mechanism and evolution characteristics of water–sand inrush caused by the instability of filling medium in karst cavity.

In profits from in predecessor's research foundation, the hydraulic sedimentary model of upstream tailings dam was established to analyze the sedimentary characteristics of tailings on the dry beach, and the unsteady seepage test was performed to investigate the effect of internal erosion on the physical and mechanical properties of tailings under heavy rainfall infiltration. The research results play an essential role in analyzing and predicting the impact of internal erosion on the stability of tailings dam.

## 2. Sedimentary Characteristics of Tailings at Different Dry Beach Slopes

The gradation, spatial distribution and porosity of tailings have an important influence on the internal erosion of tailings dam. The tailings in the upstream tailings dam show distinctive sedimentary characteristics along the dry beach slope during the discharge of tailings slurry, as shown in Figure 1. Therefore, it is necessary to analyze the sedimentary

characteristics of the upstream tailings dam before investigating the effect of internal erosion on tailings dam under heavy rainfall.

### 2.1. Experimental Materials and Model

2.1.1. Raw Tailings

The raw tailings chosen in this experiment are the waste residue after molybdenum-bismuth ore refining came from the discharge outlet of a tailings dam in southern China. The height of the tailings dam is 42.1 m with a slope ratio of 1:4, and the length of the dry beach is 50 m with a slope of 1–2%. The physical properties of the raw tailings were determined according to ASTM recommended test methods (ASTM D2216-19/D854-14/D7263-21) [18–20], as shown in Table 1.

**Table 1.** Physical properties of the raw tailings sample.

| Raw Material | Natural Density $\rho(\text{g/cm}^3)$ | Moisture Content $\omega$ (%) | Pore Ratio e | Relative Density $G_S$ | Dry Density $\rho_d(\text{g/cm}^3)$ |
|---|---|---|---|---|---|
| Tailings sample | 1.42 | 14.0 | 1.11 | 2.68 | 1.27 |

The particle characteristics of the raw tailings sample were analyzed by a screening test, and the particle size smaller than 0.075 mm was determined by a Winner2000 Laser Particle Size Analyzer [21]. The particle size distribution curve was plotted according to test results, as shown in Figure 2.

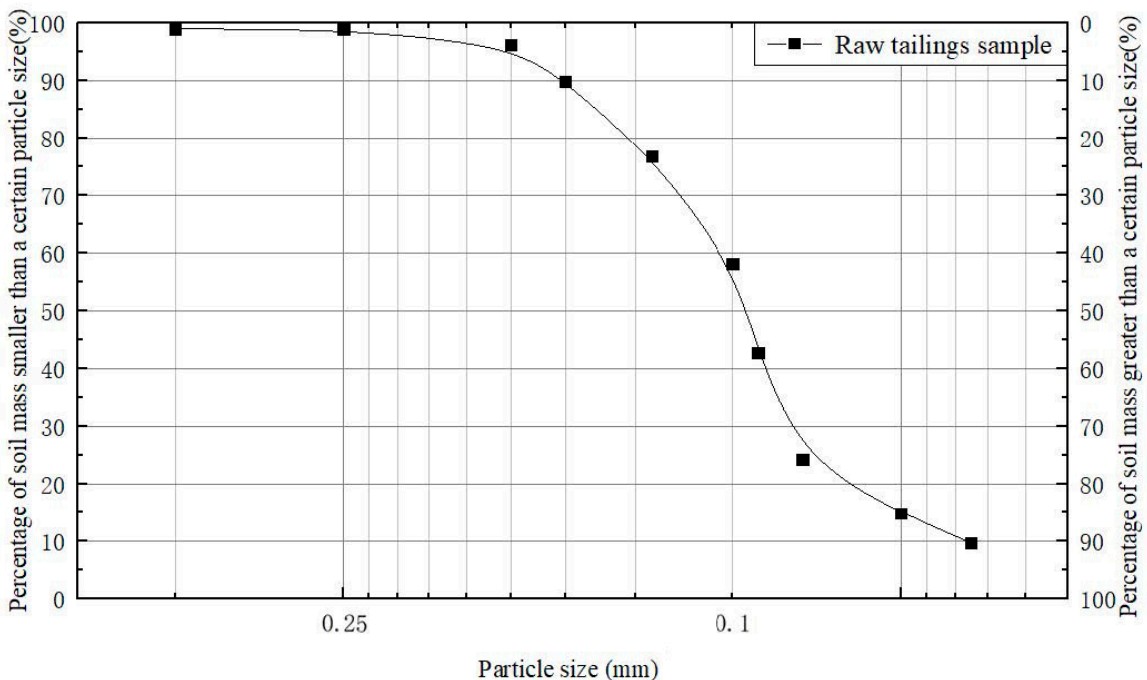

**Figure 2.** Particle size distribution curve of raw tailings sample.

The particle characteristic values of the raw tailings sample were calculated according to Figure 2, as shown in Table 2.

**Table 2.** The particle characteristics of raw tailings sample.

| Characteristic Values | Effective Particle Size $d_{10}$ (mm) | Median Particle Size $d_{30}$ (mm) | Average Particle Size $d_{50}$ (mm) | Restricted Particle Size $d_{60}$ (mm) | Non-Uniformity Coefficient $C_u$ | Curvature Coefficient $c_c$ |
|---|---|---|---|---|---|---|
| Raw tailings sample | 0.031 | 0.092 | 0.097 | 0.102 | 3.33 | 2.66 |

2.1.2. Experimental Model

A physical model for hydraulic sedimentation test of tailings was established to simulate the discharge process of tailings slurry from upstream tailings dam and to analyze the sedimentary characteristics of tailings on the dry beach. The main body of the model is a length of 500 cm plastic groove with a Trapezoidal cross-section (30 cm wide of the topline, 20 cm wide of the baseline, and 10 cm high) to simulate the dry beach of tailings dam. A height-adjustable cushion set at the upstream of the physical model was used to adjust the slope of the groove (1–2%), which is consistent with the dry beach slope of the tailings dam. A plastics discharge pipe with an inner diameter of 0.3 cm is installed at the bottom of the mixing tank to discharge the tailings slurry, as shown in Figure 3.

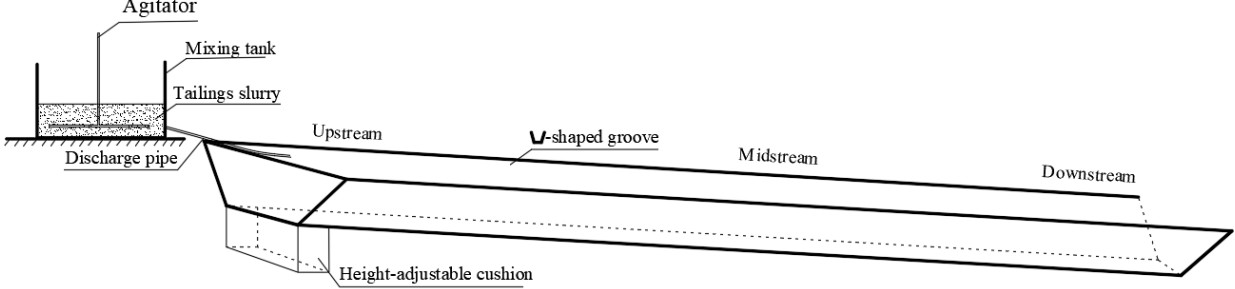

**Figure 3.** Schematic diagram of the physical model for the tailings hydraulic sedimentary test.

According to the Froude criterion [22], when fluid flow is dominated by inertial force and gravitational force, the physical model and prototype of fluid should have the same Froude number to meet the dynamic similarity conditions. The Froude number is the ratio of inertial force to gravitational force during the process of fluid flow, as shown in Formula (1):

$$F_r = \frac{v}{\sqrt{gL}} \tag{1}$$

where $F_r$ is the Froude number; $g$ is gravity; $v$ is water velocity; and $L$ is the flow length of fluid.

The flow of the tailings slurry on the dry beach is dominated by inertial force and gravitational force, so the physical model should have the same Froude number as the prototype; $F_{rm} = F_{rp}$.

$$\frac{v_m}{\sqrt{gL_m}} = \frac{v_p}{\sqrt{gL_p}} \tag{2}$$

$$\frac{L_p}{L_m} = \frac{v_p^2}{v_m^2} \tag{3}$$

where $F_{rm}$, $L_m$, $v_m$ are the Froude number, length, and water velocity of a physical model, respectively; $F_{rp}$, $L_p$, $v_p$ are the Froude number, dry beach length, and water velocity of a prototype, respectively.

The similarity ratio of the hydraulic parameters between the physical model and the prototype was derived according to the Froude criterion, as shown in Table 3.

**Table 3.** The similarity ratio of the hydraulic parameters between the physical model and prototype.

| Parameters | Similar Ratio |
| --- | --- |
| Length | 1:100 |
| Fluid flow area | 1:400 |
| Flow rate | $1:100^{1/2}$ |
| Volume flow | $1:200^{5/2}$ |

The actual flow rate of tailings slurry at the discharge outlet of the tailings dam is measured by a flow meter, which is 1.72 m/s, as shown in Figure 4.

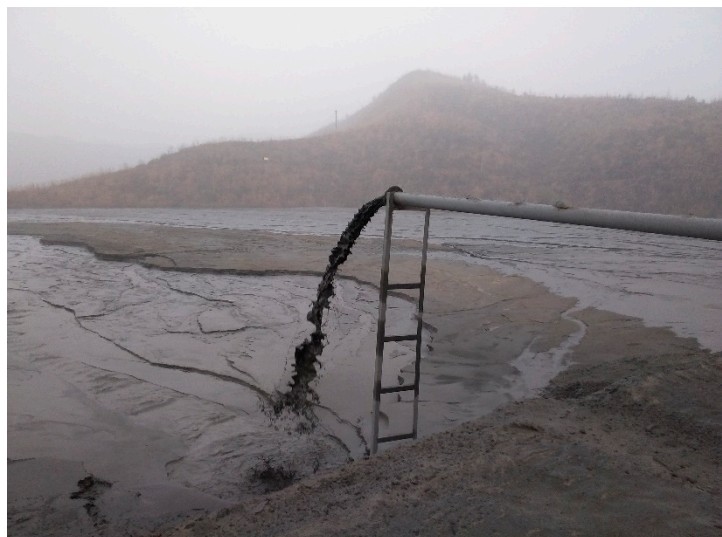

**Figure 4.** Actual flow rate of tailings slurry at the discharge outlet.

The hydraulic parameters for the physical model were calculated according to the actual flow rate and similar ratio, as shown in Table 4.

**Table 4.** The hydraulic parameters of the physical model.

| Similar Parameters | Water Velocity (cm/s) | Volume Flow (m³/s) | Mass Ratio of Water and Tailings |
|---|---|---|---|
| Test values | 17.2 | $1.21 \times 10^{-6}$ | 3:1 |

### 2.2. Experimental Procedures

(1) Adjust the height of the cushion to maintain the slope of the Trapezoidal groove at 1%.

(2) Put the tailings and water in the mixing tank at a mass ratio of 1:3 which is consistent with that of actual tailings slurry. Stir the tailings slurry 10 min to uniform, and drain it into the Trapezoidal groove through the discharge pipe at the volume flow of $1.21 \times 10^{-6}$ m³/s until the stacking height of upstream tailings exceeds 10 cm.

(3) Take the tailings samples from downstream, midstream, and upstream of the Trapezoidal groove when the water is completely drained out, numbered as sample DT, sample MT, and sample UT, respectively.

(4) Dry the tailings samples with a thermostatic drying chamber (105~110 °C) for no less than 10 h. Determine the particle size distribution of each sample by screening test and laser particle size analysis.

(5) Change the slope of the trapezoidal groove to 1.5% and 2.0%, respectively, by adjusting the height of the cushion, and repeat the procedures of step (2) to step (4).

### 2.3. Results

The particle size distribution curve of each sample is plotted according to the results of screening test and laser particle size analysis, as shown in Figure 5.

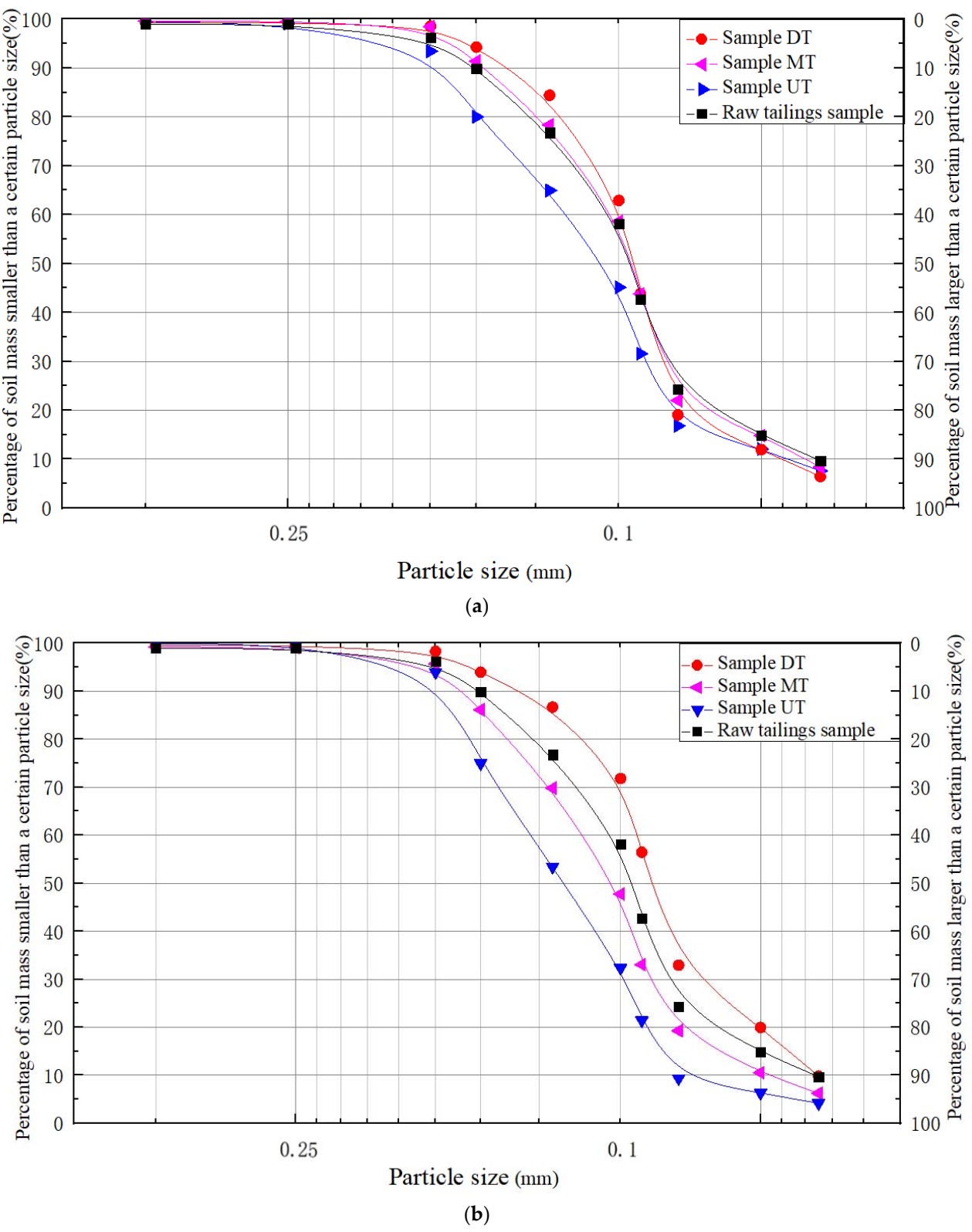

**Figure 5.** *Cont.*

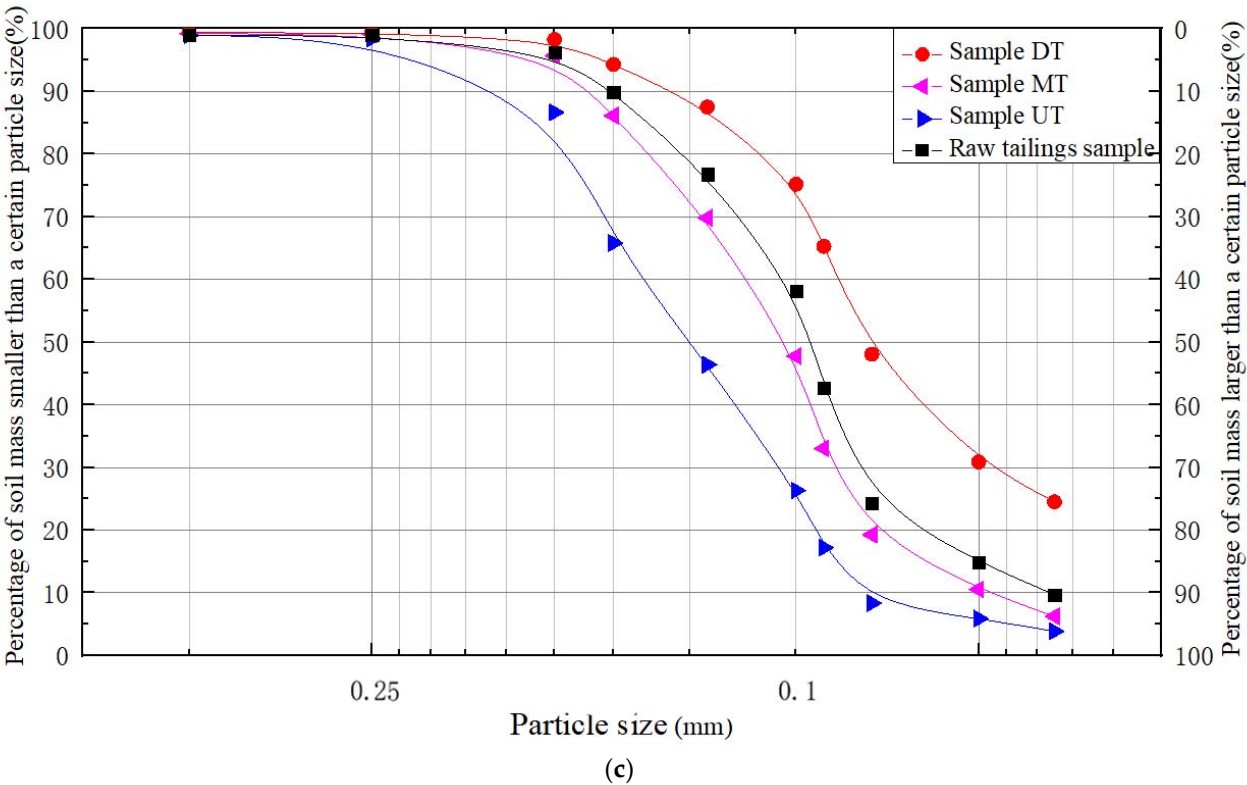

(**c**)

**Figure 5.** Particle size distribution curves of sedimentary tailings samples at different slopes. (**a**) 1% slope. (**b**) 1.5% slope. (**c**) 2% slope.

As can be seen from Figure 5, the raw tailings were stratified along the slope of the Trapezoidal groove under the hydrodynamic force and gravitational force. The slope of the Trapezoidal groove has an important effect on the sedimentary characteristics of tailings, the greater the slope, the more obvious the stratification sedimentary of tailings. When the slope is 1%, the particle size distribution curves of the samples DT, MT, and UT are shown in Figure 5a. We can see that the average particle size of each sample has no significant change compared with that of the raw tailings, indicating that a 1% slope is not conducive to the sedimentation of tailings due to low horizontal kinetic energy of tailings slurry. When the slope is 1.5%, we can see from Figure 5b that the sedimentary characteristics of tailings are more obvious than that of a 1% slope, the average particle size of the samples UT and MT increased by 16.4% and 11.4%, respectively, the average particle size of the sample DT decreased by 16.4% compared with that of the raw tailings. Additionally, the non-uniformity coefficient $C_u$ of the samples UT and MT increased due to amounts of fine-grained tailings being carried to the downstream. The amount of mid-grained tailings increased in the midstream and decreased significantly in the downstream of the model. When the slope is 2%, we can see from Figure 5c that the average particle size of the samples UT and MT have more significant increase than that of 1.5% slope. The coarse-grained tailings and fine-grained tailings are more concentrated in the upstream and downstream of the model than others, respectively, only 27.6% of the upstream tailings have particles size smaller than 0.1 mm, while more than 75.8% of the downstream tailings have particles size smaller than 0.1 mm.

*2.4. Discussion*

The flow of tailing particles is affected by hydrodynamic force and gravitational force in trapezoidal groove. As the flow velocity slows down, the hydrodynamic force decreases, and the coarse-grained tailings are deposited at first due to the greater gravity, resulting in the average particle size of upstream tailings being larger than that of downstream tailings in trapezoidal groove. With the increase in trapezoidal groove slope, the hydrodynamic

force of tailing decreases slowly, which makes the fine-grained tailings migrate farther, and has more obvious stratification sedimentary in trapezoidal groove.

According to the literature [22], the probability of tailings sedimentary law can be expressed by Formula (4):

$$\lambda = \frac{vi}{W} \qquad (4)$$

where

$\lambda$—judgment factor for tailings sedimentation,
$v$—water velocity (m/s),
$i$—hydraulic gradient, and
$W$—sedimentary velocity of tailings (m/s).

When the $\lambda$ value is greater than 1, the horizontal kinetic energy is greater than sedimentary energy, and the tailings slurry are hard to deposit; when the $\lambda$ value is less than 1, the tailings slurry are easy to deposit. According to the particle size distribution curves of the hydraulic sedimentary test, the tailings with a diameter smaller than 0.075 mm are hard to deposit and are mainly located in the downstream of the model. The tailings with a particle size of 0.075~0.15 mm are mid-grained tailings primarily located in the middle of the model, its particle size distribution along the slope of the model is the most influenced by the hydrodynamic force. The tailings with a particle diameter larger than 0.15 mm are quickly deposited on the upstream to form the tailings dam.

The average particle size distribution of sedimentary tailings along the slope of dry beach obeys the statistical law. According to the results of hydraulic sedimentary test, the average particle size $D_{50}$ as a linear function of the dry beach length $L$ at the dry beach slope of 1.5% can be expressed by Formula (5).

$$D_{50} = 0.4394 - 6.51 \times 10^{-4} \times L \qquad (5)$$

The relationship of average particle size $D_{50}$ with the distance of sampling point from upstream $L$ is shown in Figure 6. We can see that the test value of average particle size was consistent with that of on-site measured values, indicating that the results of hydraulic sedimentary test can accurately reflect the sedimentary characteristics of dry beach tailings.

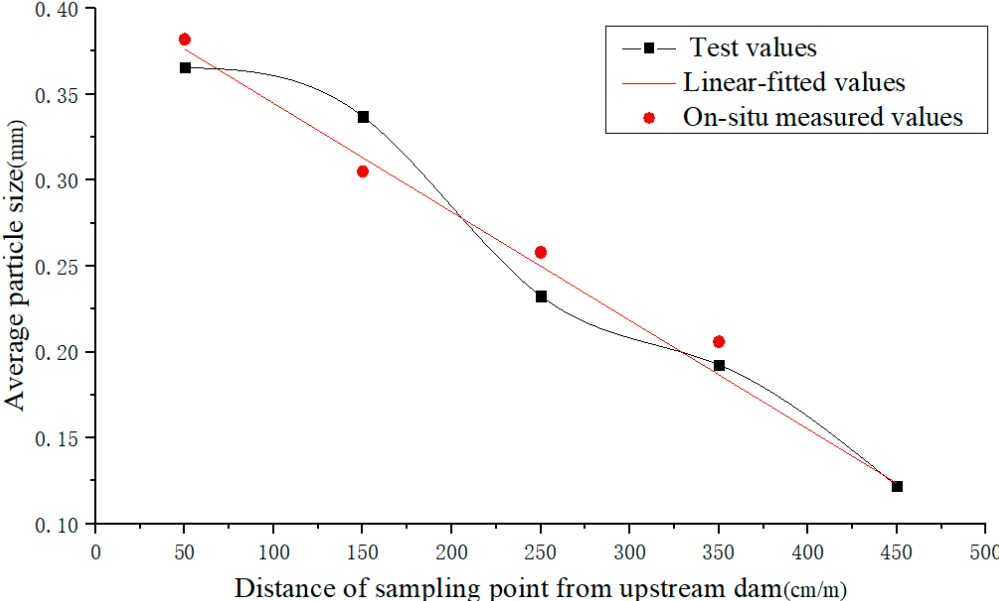

**Figure 6.** The relationship between average particle size $D_{50}$ and the distance of sampling point from upstream dam $L$.

### 3. The Effect of Internal Erosion on the Physical and Mechanical Properties of Tailings under Heavy Rainfall Infiltration

*3.1. Critical Condition of Internal Erosion*

In the porous medium, the fine-grained tailings are easy to be washed away under the seepage force, and the derivation process of the critical hydraulic gradient of internal erosion is derived as follows [23]:

The weight of saturated fine-grained tailings $G'$:

$$G' = (\gamma_s - \gamma_w) \cdot V \tag{6}$$

where

$G'$—Weight of fine-grained tailings (N),
$\gamma_s$—Bulk unit weight of fine tailings (N/m$^3$),
$\gamma_w$—Bulk unit weight of water (N/m$^3$), and
$V$—Volume of fine tailings.

The seepage force of fine-grained tailings is $F$:

$$F = i\gamma_w \cdot V \tag{7}$$

The initiation of fine-grained tailings should overcome the static friction force between the particles due to the cohesion of saturated tailings is 0 [24,25]:

$$f = G' \cdot tg\varphi \tag{8}$$

where

$f$—the static frictional resistance between the fine-grained tailings (N), and
$\varphi$—internal friction angle of underwater fine-grained tailings (°).

The critical equilibrium condition for fine-grained tailings initiation is $F = f$, so the critical hydraulic gradient of internal erosion is $i_{cr}$:

$$i_{cr} = \frac{G' \cdot tg\varphi}{\gamma_w \cdot V} \tag{9}$$

*3.2. Experimental Model*

A sand columns model was established to investigate the effect of internal erosion on the physical and mechanical properties of tailings under unsteady seepage caused by heavy rainfall infiltration, as shown in Figure 7. The main body of the physical model consists of three sand columns (Column 1, Column 2, and Column 3), which are connected by flanges. Additionally, each sand column is made of polyvinyl chloride (PVC) with a maximum internal diameter 15 cm and a length of 30 cm. The water pressure gauges are installed on the top of the model at a distance of 15 cm to measure the water head differences of each sand Column, which can be controlled by adjusting the flow rate of the outlet flowmeter. A water tank with a maximum height of 3 m connect to the model is used to simulate the raising saturation line of tailings dam under heavy rainfall infiltration.

*3.3. Experimental Procedures*

(1) Put the sedimentary tailings into the corresponding sand columns. Fill the samples DT, MT, and UT in the hydraulic sedimentary test into Column 1, Column 2, and Column 3, respectively, to make the seepage direction of the physical model consistent with the actual seepage direction of the tailings dam.

(2) After installing the physical model, open the inlet valve to saturate the sand columns for 24 h with water.

(3) Then, adjust the outlet flowmeter to control the water head differences of each sand Column. When the value of each water pressure gauge is stable, record the flow Q and water head differences of each sand column.

(4) Raise the water level of the water tank at a rate of 0.1 m/h for 24 h, which is consistent with the rising rate of saturation line of the tailings dam under heavy rainfall infiltration in a 50 years return period. Record the flow volume and the water head differences between the water pressure gauges each 1 h.

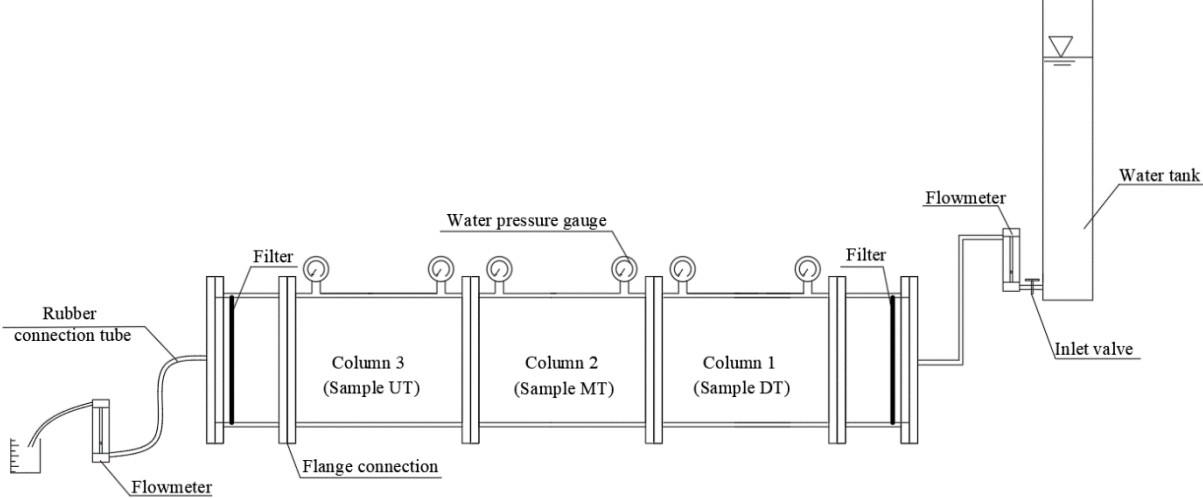

**Figure 7.** Schematic diagram of unsteady seepage model.

### 3.4. Results

#### 3.4.1. The Effect of Internal Erosion on the Hydraulic Conductivity of Tailings Samples

The hydraulic conductivity K of each sand column is calculated according to flow Q and water head differences. The effect of internal erosion on the hydraulic conductivity of each sand column under unsteady seepage is shown in Figure 8.

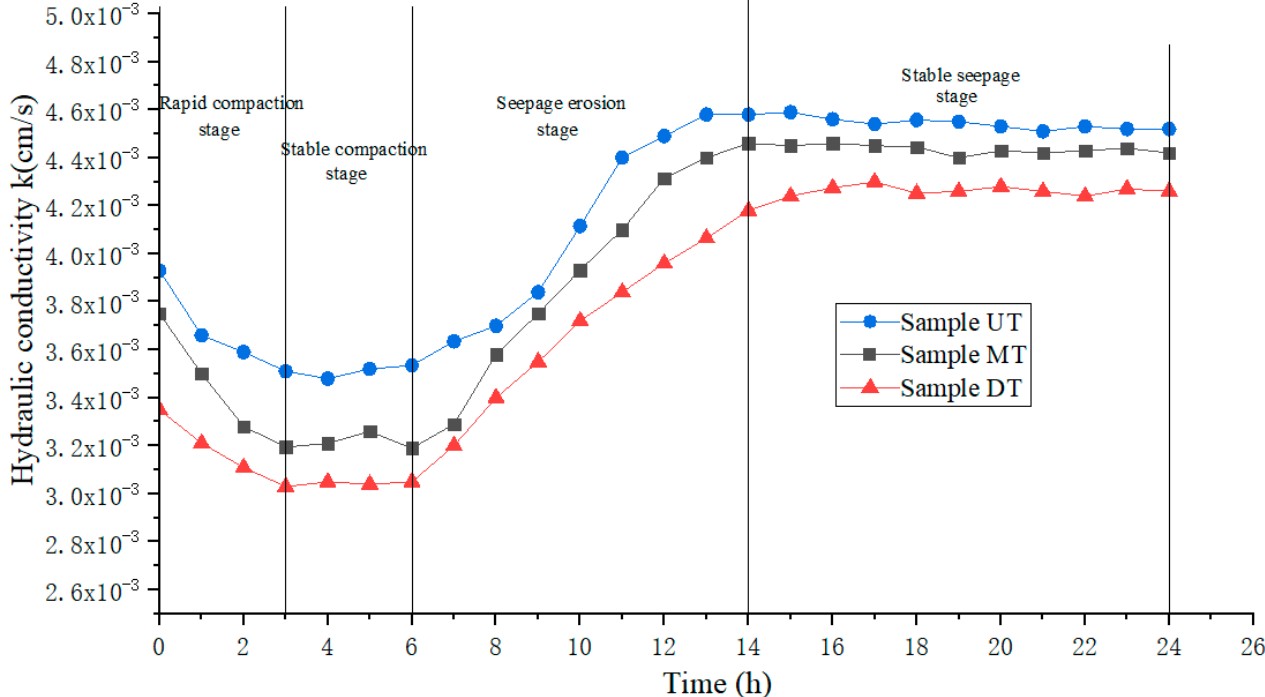

**Figure 8.** Changes in hydraulic conductivity with time under unsteady seepage.

As can be seen from Figure 8, initial hydraulic conductivity of each sand column is $3.35 \times 10^{-3}$ cm/s (sample DT), $3.75 \times 10^{-3}$ cm/s (sample MT), and $3.93 \times 10^{-3}$ cm/s (sample UT). After 24 h, the hydraulic conductivity of three sand columns increases by 27.2, 17.9, and 15.3%, respectively (i.e., $4.26 \times 10^{-3}$, $4.42 \times 10^{-3}$, and $4.53 \times 10^{-3}$ cm/s).

The change in hydraulic conductivity with time of sand columns can divide into four stages: rapid compaction stage, stable compaction stage, internal erosion stage, and stable seepage stage. In stage 1 (1–3 h), there was the maximum decrease in hydraulic conductivity; the hydraulic conductivity of the samples DT, MT, and UT reduced by 9.5%, 14.9%, and 10.7%, respectively. The fine-grained tailings have a small range of migration due to the sudden increase in seepage force, which reduces the pore volume of downstream tailings and the hydraulic conductivity. In stage 2 (3–6 h), the hydraulic conductivity of each sand column tends to be stable due to the migration of fine-grained tailings and the increase in water head contributes to the balance of the hydraulic conductivity. In stage 3 (6–14 h), there was the maximum increase in hydraulic conductivity, and the hydraulic conductivity of the samples DT, MT, and UT increased by 37.1%, 39.8%, and 29.7%, respectively. In stage 4 (14–24 h), the hydraulic conductivity of tailings samples remains constant regardless of the increase in water head, and the tailings achieved a new stable state of hydraulic conductivity due to the particles of tailings being reorganized under the seepage force.

### 3.4.2. The Effect of Internal Erosion on the Shear Strength of Tailings Samples

The tailings samples have been taken from each sand column in situ for direct shear test [26] after internal erosion test. The control shear rate was 0.8 mm/min. The shear strength and internal friction angle of each tailings sample before and after internal erosion are shown in Table 5.

**Table 5.** The shear strength of tailings samples before and after internal erosion.

| Time | Tailings Samples | Reduction Rate of Shear Strength (%) | Internal Friction Angle (°) |
|---|---|---|---|
| Before internal erosion | Sample DT | | 35.14 |
| | Sample MT | | 36.69 |
| | Sample UT | | 38.70 |
| After internal erosion | Sample DT | 20.9 | 30.21 |
| | Sample MT | 15.1 | 32.94 |
| | Sample UT | 12.4 | 35.48 |

As can be seen from Table 5, the shear strength and internal friction angle of the tailings samples decreased after internal erosion, the shear strength of sample DT (Column 1), sample MT (Column 2), and sample UT (Column 3) decreased by 20.9%, 15.1%, and 12.4%, respectively, under 100 kPa normal press. The internal friction angle of the samples DT, MT, and UT decreased 14.0%, 10.2%, and 8.3%, respectively.

### 3.4.3. The Effect of Internal Erosion on the Particle Size Distribution Curves of Tailings Samples

To further analyze the effect of internal erosion on the physical properties of the tailings samples, the samples were taken for a screening test after internal erosion. The comparison of particle-size distribution curves of tailings samples before and after internal erosion test is shown in Figure 9.

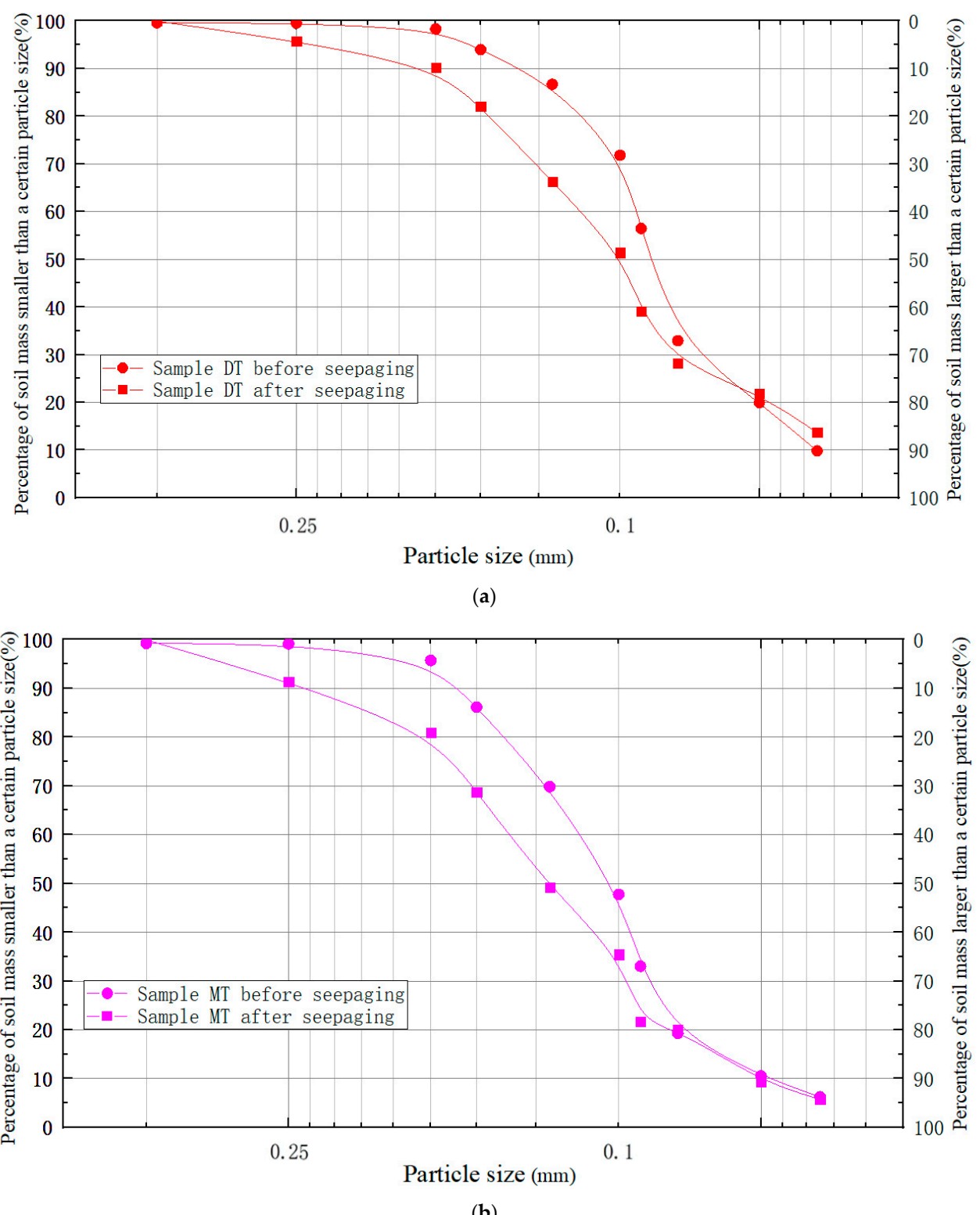

(**a**)

(**b**)

**Figure 9.** *Cont.*

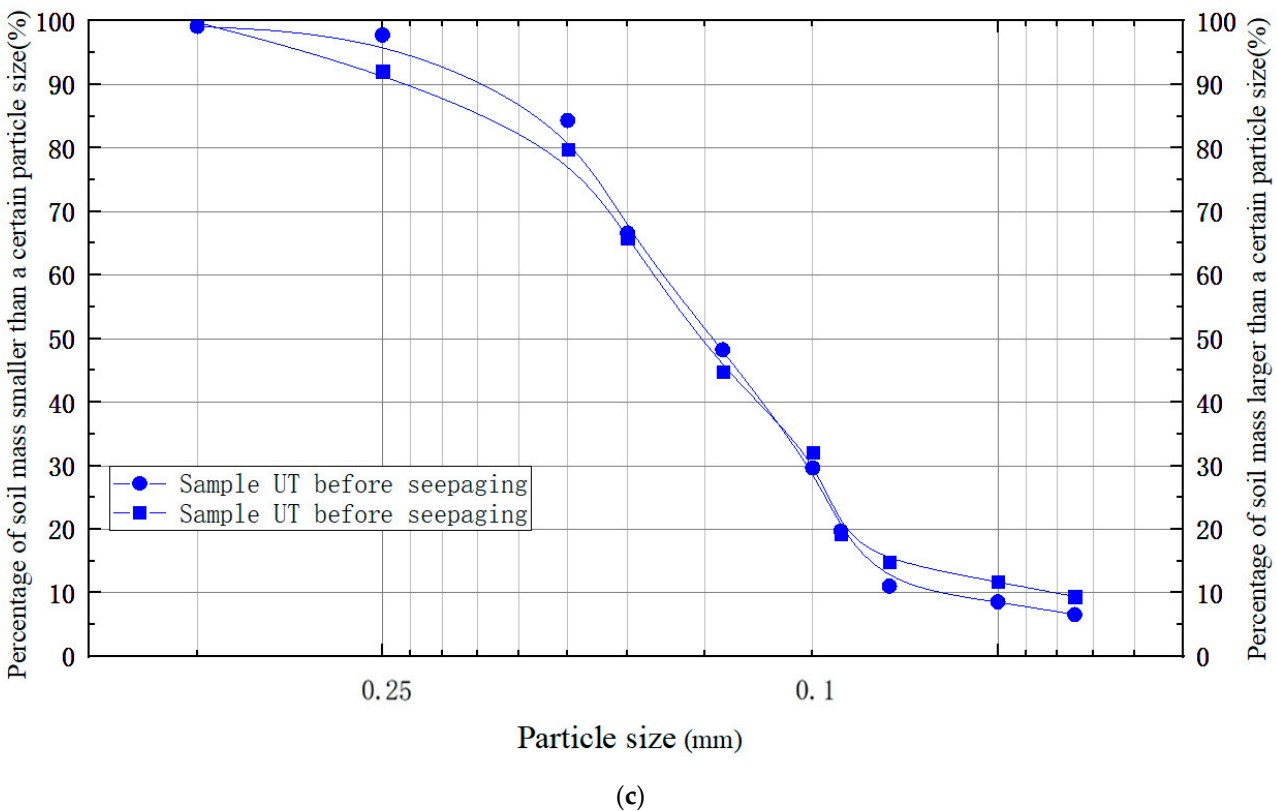

(**c**)

**Figure 9.** Comparison of particle-size distribution curves of tailings samples before and after internal erosion. (**a**) Sample DT. (**b**) Sample MT. (**c**) Sample UT.

As can be seen from Figure 9, the average particle size of sample DT (Column 1), sample MT (Column 2), and sample UT (Column 3) is increased by 6.4%, 12.0%, and 2.4%, respectively, due to the migration of the fine-grained tailings along the seepage direction. About 20% of tailings with particle size smaller than 0.075 mm and 10% tailings with the size of 0.075~0.1 mm migrated to the downstream in the sample DT according to Figure 9a. The change in average particle size of each tailings sample is not obvious; however, the mass of tailings with the particle size below 0.1 mm increased by about 4% in sample UT according to Figure 9c.

*3.5. Discussion*

It can be seen from the internal corrosion test that the permeability of tailings has changed significantly under unsteady seepage caused by heavy rainfall infiltration. When the continuously rising water head gradient reaches the critical head gradient of tailings, lots of fine-grained tailings are washed away to the downstream, resulting in internal erosion and changing the skeleton structure of tailings. The critical hydraulic gradients of the samples DT, MT and UT are calculated according to Formula (9), which are 0.32, 0.36, and 0.41, respectively.

The result of a direct shear test showed that internal erosion has the greatest influence on the shear strength and internal friction angle of fine-grained tailings compared to the coarse-grained tailings. The main reason for this phenomenon is the large-scale migration of fine particles in the upstream tailings, which leads to the change of particle size distribution of tailings. This conclusion is also confirmed in Figure 9.

The upstream tailings dam is more susceptible to internal erosion under unsteady seepage caused by heavy rainfall infiltration. The fine-grained tailings will be carried out of the tailings dam by the water, which is consistent with the actual observations. If this phenomenon is not seriously concerned, the safety of tailings dam will be affected over time.



## 4. Conclusions

Based on the above analysis and discussion, the following conclusions can be drawn:

(1) The average particle size $D_{50}$ of sedimentary tailings decreases along the slope of dry beach. The beach slope has an important effect on the sedimentary characteristics of tailings, the greater the slope, the more obvious the stratification sedimentary of tailings.

(2) The average particle size distribution along the groove slope in the hydraulic sedimentary test was consistent with that of on-site measured values, indicating that the test results can accurately reflect the sedimentary characteristics of dry beach tailings.

(3) The critical hydraulic gradient of the samples DT, MT, and UT are 0.32, 0.36, 0.41, respectively. When the hydraulic gradient of each tailing sample exceeds the critical value, the migration of the fine-grained tailings will result in internal erosion.

(4) The migration of fine-grained tailings caused by internal erosion increases the tailings' permeability and reduces the tailings' shear strength. After internal erosion, the hydraulic conductivity of the samples DT, MT, and UT increased by 27.2%, 17.9%, and 15.3%, respectively, and the shear strength of each samples decreased by 20.9%, 15.1%, and 12.4%, respectively.

(5) The average particle size of the samples DT, MT, and UT increased by 6.4%, 12.0%, and 2.4%, respectively, due to the migration of the fine-grained tailings along the seepage direction. More than 20% fine-grained tailings in the sample DT migrated downstream.

**Author Contributions:** Conceptualization, R.G. and G.H.; Investigation, R.G.; Project administration, G.H.; writing—original draft, R.G.; writing—review & editing, G.H. funding acquisition, G.H. Both authors have read and agreed to the published version of the manuscript.

**Funding:** This research was funded by [National Natural Science Foundation of China] grant number [51804164, 51974163], [Natural Science Foundation of Hunan Province] grant number [2019JJ50496, 2021JJ30571], [Science and Technology Department Key R&D Plan Project of Hunan Province] grant number [2017SK2280].

**Institutional Review Board Statement:** Not applicable.

**Informed Consent Statement:** Not applicable.

**Data Availability Statement:** Some or all data, models, or code that support the findings of this study are available from the corresponding author upon reasonable request.

**Acknowledgments:** This study was also supported by Hunan Province Engineering Research Center of Radioactive Control Technology in Uranium Mining and Metallurgy & Hunan Province Engineering Technology Research Center of Uranium Tailings Treatment Technology, University of South China.

**Conflicts of Interest:** The authors declare no conflict of interest.

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
