# Peer review of "The Effects of Internal Erosion on the Physical and Mechanical Properties of Tailings under Heavy Rainfall Infiltration"

_applsci, doi:10.3390/app11209496_

Round 1
Reviewer 1 Report
The paper presents an analysis of the effects of internal erosion on tailings' physical and mechanical properties under heavy rainfall infiltration. Mining tailings behavior analyzes are essential for the development of safe tailings dam designs. The paper presents an experimental campaign to evaluate the tailings sedimentation process and evaluate the influence of the flow on material properties; however, some gaps need to be verified, as presented below:
- There are some deficiencies in the punctuation of the text Introduction. Some examples:
Introduction, Second paragraph:
"…indicated that heavy rainfall was the critical factor that contributes to 30% of the dam-failure accidents[5-7]. when the…" (check the punctuation);
Pg. 4 first and fourth paragraph: check the final punctuation.
Please check all the text.
- Particle-size distribution curves: Please check the shaft scale corresponding to the grain diameter. Use whole numbers (0.01, 0.05, etc.) and an enlarged scale. Please replace the use of "/" for parentheses or square brackets to indicate units. Thus, there is no confusion with the arithmetic function of division;
- The introduction should better explain the model of internal erosion in upstream tailings dams caused by heavy rains. How would be generated the charge difference for the flow, in Figure 1? Could the sedimentation process explored in item 2.1 not be affected by the operational activity of each dam? When someone reads item 2.1, it is not clear the purpose of this experiment in evaluating internal erosion. Would you mind explaining in the prior text?
- What was the ore that generated the analyzed tailings?
- The studied waste does not present plasticity? Have tests been carried out? This is essential information for erosion analysis.
- Is mining tailings a granular material? Does the analysis of the values ​​of Cu and Cc make sense in this type of material, which appears to be relatively fine?
- Item 2.2: Please explain the 1:3 ratio adopted for water and tailing and volume flow.
- Pg. 9, line 6: the cohesion of saturated tailings equal to 0 was obtained by tests? Please explain. Check the numeration of Equation "89".
- General question: the internal erosion process evaluated by the authors regarding unsteady seepage would be related to the possibility of static liquefaction occurring in low-density granular tailings?
Author Response
- There are some deficiencies in the punctuation of the text Introduction. Some examples:
Introduction, Second paragraph:
"…indicated that heavy rainfall was the critical factor that contributes to 30% of the dam-failure accidents[5-7]. when the…" (check the punctuation);
Pg. 4 first and fourth paragraph: check the final punctuation.
Author’s Response:
We carefully checked the English language and style of the manuscript, including the punctuation.
Please check all the text.
- Particle-size distribution curves: Please check the shaft scale corresponding to the grain diameter. Use whole numbers (0.01, 0.05, etc.) and an enlarged scale. Please replace the use of "/" for parentheses or square brackets to indicate units. Thus, there is no confusion with the arithmetic function of division;
Author’s Response:
Checked.And we replace the use of "/" for parentheses or square brackets in Fig.2, Fig.5, Fig.6, Fig.9.
- The introduction should better explain the model of internal erosion in upstream tailings dams caused by heavy rains. How would be generated the charge difference for the flow, in Figure 1? Could the sedimentation process explored in item 2.1 not be affected by the operational activity of each dam? When someone reads item 2.1, it is not clear the purpose of this experiment in evaluating internal erosion. Would you mind explaining in the prior text?
Author’s Response:
(1) According to your suggestion,we added a diagram of the rise of the saturation line under heavy rain in Fig.1, and analyze the changes of water pressure with the rise of the saturation line in the introduction.
(2)The operational activity of each dam will affect the distribution of tailings in tailings pond, but there will be the same sedimentation law of tailings along the dry slope direction during the discharge of tailing slurry, that is, the upstream tailings are coarse, and the downstream tailings are fine, and the stratification characteristics of tailings are related to the slope.
(3) We added and explained why upstream tailings dams are more prone to seepage erosion under heavy rainfall in the introduction.
- What was the ore that generated the analyzed tailings?
Author’s Response:
The main components of the ore that generated the analyzed tailings are molybdenum and bismuth.
- The studied waste does not present plasticity? Have tests been carried out? This is essential information for erosion analysis.
Author’s Response:
Considering that the tailings is a kind of sandy soil, we did not analyze the plasticity of tailings. Thank you very much for your suggestions. We will carry out the research on this part in the future.
- Is mining tailings a granular material? Does the analysis of the values ​​of Cu and Cc make sense in this type of material, which appears to be relatively fine?
Author’s Response:
Mining tailings is a graded granular material. We believe that the values ​​of Cu and Cc are still relevant when analyzing the sedimentary law and erosion characteristics of tailings.
- Item 2.2: Please explain the 1:3 ratio adopted for water and tailing and volume flow.
Author’s Response:
The mass ratio of water and tailings in the physical model is consistent with that of actual tailing slurry. The volume flow in the physical model is determined by actual volume flow and Similar ratio.
- Pg. 9, line 6: the cohesion of saturated tailings equal to 0 was obtained by tests? Please explain. Check the numeration of Equation "89".
Author’s Response:
The cohesion of saturated tailings equal to 0 is determined by relevant literature and engineering experience. The cohesion of saturated sandy soil is 0.
The numeration of Equation should be “(8)”. It has been revised in the manuscript.
- General question: the internal erosion process evaluated by the authors regarding unsteady seepage would be related to the possibility of static liquefaction occurring in low-density granular tailings?
Author’s Response:
Saturated low-density tailings at low-confining pressure are more susceptible to static liquefaction under the seepage force,but no static liquefaction of tailings occurred in the unsteady seepage test due to no significant change in shear strength after internal erosion.It may be related to the physical model of unsteady seepage test. Dynamic liquefaction and static liquefaction of tailings are also the major research area of our research group. You are welcome to provide more opinions and suggestions about this topic.

Reviewer 2 Report
Dear Authors,
The proposed analysis and the study is quite interesting, but the overall quality of the presentation is scarce.
Sections and paragraphs are not correctly numbered, in addition to a poor-described method section and a scarcely organized description of the results. These last are integrated into the discussion, preventing an easy reading.
First of all, I suggest improving and completing methods, then describing only the results obtained and finally proposing speculations and interpretations.
Best Regards

Author Response
Dear Pro.
Thank you for your consideration of this manuscript we submitted and the suggestions put forward by you. We have made serious revisions to the manuscript's contents and English language according to your suggestions, see revised version for details. If you have any questions about this manuscript, please do not hesitate to contact me at any time.
Yours Sincerely,
Rong Gui
Email:guirong606@gmail.com

Round 2
Reviewer 1 Report
The authors presented the answers to my questions. However, some of the aspects should be also included in the paper text to help the understanding.
For example, it will be interesting to cite on paper the ore that originated the tailing analyzed and explain the 1:3 ratio adopted for water and tailing and volume flow. Would you please mention in the text the literature that embased the adoption of cohesion (or cohesion intercept) equal to zero for tailings?
Author Response
Dear Pro.
According to your suggestion, we added the ore type of tailings before refining in section 2.1.1,and explained the reason of the mass ratio of tailings to water we adopted in the physical model in section 2.2(2).
We added two literatures([24] and [25]) to support the statement that the cohesion of saturated tailings equal to 0.
Thank you very much for your positive evaluation of the manuscript. Your suggestions are very helpful to improve the quality of the manuscript.
Yours Sincerely,
Gui Rong

Reviewer 2 Report
Dear Authors,
thank you for improving your manuscript, which I find more complete and clear. I would suggest an additional minor revision, essentially regarding text spacing and improvements for the symbols in the figures (i.e., fig. 1, 5, 8 and 9).
All suggestions are reported in the comments of the attached file.
Best regards

Author Response
Dear Pro.
Thank you very much for your suggestions and positive evaluation of the manuscript. We have made a comprehensive revision of the manuscript according to your suggestion.Your suggestions are very helpful to improve the quality of the manuscript.
Yours Sincerely,
Gui Rong

This manuscript is a resubmission of an earlier submission. The following is a list of the peer review reports and author responses from that submission.